# MtDNA D-Loop Diversity in Alpine Cattle during the Bronze Age

José Granado [1], Marianna Harmath [1], Umberto Tecchiati [2], Klaus Oeggl [3], Jörg Schibler [1] and Angela Schlumbaum [1,*]

1 Integrative Prehistoric and Archaeological Science, Department Environmental Sciences, University of Basel, Spalenring 145, CH 4055 Basel, Switzerland; jose.granado@unibas.ch (J.G.); marianna.harmath@gmail.com (M.H.); joerg.schibler@unibas.ch (J.S.)

2 Laboratorio di Preistoria, Protostoria ed Ecologia Preistorica, Dipartimento di Beni Culturali e Ambientali, Università Degli Studi di Milano, PrEcLab, Via Noto 6, I-20122 Milano, Italy; umberto.tecchiati@unimi.it

3 Institut für Botanik, Universität Innsbruck, Sternwartestr. 15, A-6020 Innsbruck, Austria; klaus.oeggl@uibk.ac.at

* Correspondence: angela.schlumbaum@unibas.ch; Tel.: +41-61-2074218

**Abstract:** The Bronze Age in Europe is characterized by major socio-economic changes, including certain aspects of animal husbandry. In the Alpine region archaeozoological data, though not very abundant, reveal that cattle were the most important domestic animals in this time period. They were probably used differently in the lowlands than at higher altitude, traction became more important and people increasingly exploited them for dairy products rather than for meat. Thus, a crucial question in this context is whether these major events are accompanied by changes in genetic diversity of cattle. Here we report partial mtDNA d-loop data (320 bp) obtained by PCR from 40 alpine cattle excavated at different sites in South Tyrol, Italy, and Grisons, Switzerland. Most cattle belong to the main European taurine T3 haplogroup, but a few members of T2 and Q haplogroups were identified. Moreover, genetic diversity measures and population genetic statistics indicate different cattle histories at different sites, including bottlenecks and potential admixture. However, Bronze Age Alpine cattle appear to be linked to modern rural cattle mainly from Italy.

**Keywords:** *Bos taurus*; ancient DNA; archaeozoology; Switzerland; South Tyrol; genetic diversity; population

## 1. Introduction

Europe underwent substantial socio-economic and cultural changes during the Bronze Age [1,2]. Here, metallurgy played a formative role, stimulating technological innovations and triggering socio-economic transformations which were enhanced by extensive human migrations [3] and extended trade networks [4]. Furthermore, this development affected agriculture: different crop plants (e.g., millets, spelt, lentil) and domestic animals (e.g., horse) [5,6] appeared compared to earlier time periods. During this time, the Alps, which previously seemed to be a hostile environment for human settlement, began to be populated, mainly in connection with the exploitation of the abundant ore deposits in several regions of the Alpine arch. New settlements were founded both in the valley bottoms and at high altitudes. In the wake of mining activities, human impact on the landscape increased and the intensification of animal husbandry expanded into marginal areas of the Alps, e.g., [7–9].

It is however still a matter of debate whether the exploitation of high altitudes was accompanied by alpine farming, in particular with increased seasonal vertical transhumance in form of pastoralism on alpine grassland, e.g., [10]. This debate has been fueled by palynological studies in the territory of the Neolithic glacier mummy "Ötzi", which place the onset of alpine farming already in the Neolithic [11,12]. However, combining

archaeological and palynological data, farming in the high-altitudes of the Alps started with the Bronze Age or slightly before [10].

Archaeozoological research in some regions of the Alpine arc, such as the Etsch River (ital. Adige) basin in South Tyrol, Italy, and in canton Grisons, Switzerland [13], revealed that during the Bronze Age, cattle size decreased [9,14] at different paces. The smaller cattle which appeared indicate an introduction of a different population [13,15]. Additionally, slaughter age increased, e.g., [14], supporting a change in use of meat in favor of secondary products such as milk and as draught animals [8,10,13,14,16–21]. Unsurprisingly, because of the multitude of different environments, these developments are not observed at all sites at the same time. It is worth mentioning that lactose tolerance started in the Balkans [22], but the tolerance allele was not frequent during the Bronze Age [3].

Concerning cattle, the pivotal questions are (i) were cattle specialized for grazing on alpine meadows?; (ii) were cattle used as meat or milk suppliers, or were they also used for transport?; (iii) were these cattle locally adapted or imported from other regions for this purposes?

One way of tackling these questions is to investigate genetic diversity during these time periods using ancient DNA techniques. Given new technologies in ancient DNA such as whole genome sequencing or shotgun approaches, it was nonetheless shown, that powerful information about past animal husbandry practices can be gained from single loci such as mtDNA and even from short d-loop fragments obtained from archaeological samples, e.g., [23–25]. Taurine cattle (*Bos taurus*) were domesticated about 9000 BCE in the Near East and subsequently spread into Europe with Neolithization [26], reaching the Eastern Alpine arc via the Balkans and Danube around the mid-6th millennium cal BCE [27,28]. Genetically, these early cattle reflect Near East mtDNA haplotypes with predominance of members of the T3 haplogroup, but also T2, T1, and Q as seen in other regions during the Neolithic, e.g., [26,29]. The further genetic history of cattle in the Alps/Central Europe is poorly known. What exactly the genetic background in the alpine area was is still unknown and what happened genetically after the initial introduction of cattle is unclear.

Here we sequenced 320 bp of the mtDNA d-loop from domestic cattle excavated at alpine metallurgical sites in the Eisack valley (South Tyrol, North Italy) and from Savognin Padnal, canton Grisons (Switzerland). The aim was to determine maternal lineage composition and diversity in Bronze Age alpine cattle, and to explore whether shifts in cattle use and husbandry during the Bronze Age are preceded or accompanied by changes in mitochondrial d-loop diversity in the alpine region. We also asked whether there is a legacy of maternal lineages of alpine Bronze Age cattle into extant native alpine cattle breeds (genetic continuity?).

We conclude that different cattle histories are reflected at three different sites and that genetic links to modern cattle breeds, mainly from Italy, have survived.

## 2. Materials and Methods

### 2.1. Samples for Archaeogenetic Analyses

Cattle bone and teeth samples were selected from the Late Chalcolithic sites of Klausen-Gufidaun-Plank, Early Bronze Age of Vahrn Nössingbühel, Middle Bronze Age of Brixen Albanbühel, and Late Bronze Age of Brixen Köstlanerstrasse, in South Tirol, Italy (Figure 1). Cal BCE 2 sigma dates (conducted by the Klaus-Tschira-Archäometry-Center, Curt-Engelhorn-Center Archäometrie gGmbH, Mannheim, Germany) are given in (Table 1). In addition, teeth and horn core samples were collected from Savognin-Padnal in canton Grisons, Switzerland (Table 1). These samples are dated by AMS $^{14}$C (CEP, University of Bern) and according to typology (Table 1) [14,19,30].

**Table 1.** Details of cattle samples and haplogroup identification. [14]C datings from Savognin Padnal were taken from Bopp-Ito, M. Animal husbandry in the Bronze Age Alpine settlement "Savognin Padnal", Switzerland: a preliminary study. In Proceedings of the General Session of the 11th International Council for Archaeozoology Conference (Paris, 23–28 August 2010), Levèvre, C., Ed. BAR International Series 2012; Vol. 2354, pp. 75–85. BA = Bronze Age; - = no dating.

| | Sample Code | Genbank Acc. No./Lab Code | Archaeological Code/Horizon | Site | Element | Dating | 14 C Lab Code | Dating | MtDNA Haplogroup |
|---|---|---|---|---|---|---|---|---|---|
| | | | | | | cal BCE 2 sigma | | typologically/cultural period | |
| 1 | NOS1 | MT423754.1 | 495 | Vahrn, Nössingbühel | M3 | 1949–1776 | MAMS-23501 | Early BA | T3 |
| 2 | NOS2 | MT423755.1 | 1146 | Vahrn, Nössingbühel | M3 | 1737–1541 | MAMS-23502 | Early BA | T3 |
| 3 | NOS3 | MT423756.1 | 1212 | Vahrn, Nössingbühel | M3 | 1875–1687 | MAMS-23503 | Early BA | T3 |
| 4 | NOS4 | MT423757.1 | 376 | Vahrn, Nössingbühel | M3 | 2111–1915 | MAMS-23504 | Early BA | T3 |
| 5 | NOS5 | OEG5 | 1044 | Vahrn, Nössingbühel | M3 | neg | | Early BA | neg |
| 6 | NOS6 | OEG6 | 407 | Vahrn, Nössingbühel | left jawbone with 3 molars | - | | Early BA | neg |
| 7 | NOS7 | MT423758.1 | 470 | Vahrn, Nössingbühel | jawbone with 3 molars | - | | Early BA | T3 |
| 8 | ALB8 | MT423741.1 | M3406 | Brixen, Albanbühel | maxillare | 1602–1438 | MAMS-23506 | | T3 |
| 9 | ALB9 | MT423742.1 | A9717 | Brixen, Albanbühel | maxillare | 1683–1531 | MAMS-23507 | | T3 |
| 10 | ALB10 | MT423767.1 | A3744 | Brixen, Albanbühel | molar | 1863–1642 | MAMS-23508 | | T3 |
| 11 | ALB11 | OEG11 | A5500 | Brixen, Albanbühel | molar | 1613–1502 | MAMS-23509 | | neg |
| 12 | ALB12 | MT423768.1 | A10130 | Brixen, Albanbühel | molar | 1496–1418 | MAMS-23510 | | T3 |
| 13 | ALB13 | OEG13 | A8568 | Brixen, Albanbühel | molar | 1681–1531 | MAMS-23511 | | neg |
| 14 | ALB14 | OEG14 | A7071 | Brixen, Albanbühel | molar | 1732–1533 | MAMS-23512 | | neg |
| 15 | ALB15 | MT423744.1 | 13056 | Brixen, Albanbühel | jawbone with 2 molars | 1518–1424 | MAMS-23513 | | T3 |
| 16 | ALB16 | MT423769.1 | 15278 | Brixen, Albanbühel | jawbone with 2 molars | 1260–1055 | MAMS-23514 | | T3 |
| 17 | ALB25 | OEG25 | 16419 | Brixen, Albanbühel | metatarsus distal | - | | Early BA | neg |
| 18 | ALB26 | MT423743.1 | 13511 | Brixen, Albanbühel | metatarsus proximal | - | | Early BA | T3 |

**Table 1.** *Cont.*

| | Sample Code | Genbank Acc. No./Lab Code | Archaeological Code/Horizon | Site | Element | Dating | 14 C Lab Code | Dating | MtDNA Haplogroup |
|---|---|---|---|---|---|---|---|---|---|
| 19 | ALB27 | MT423736.1 | 13684 | Brixen, Albanbühel | metarsus distal | - | | Early BA | T2 |
| 20 | ALB28 | MT423739.1 | 14596 | Brixen, Albanbühel | metacarpus proximal | - | | Early BA | T3 |
| 21 | ALB29 | MT423770.1 | 15722 | Brixen, Albanbühel | metatarsus proximal | - | | Early BA | T3 |
| 22 | ALB30 | MT423737.1 | 15729 | Brixen, Albanbühel | metatarsus distal | - | | Early BA | T2 |
| 23 | ALB31 | MT423738.1 | 13611 | Brixen, Albanbühel | metatarsus mitte | - | | Early BA | T3 |
| 24 | ALB32 | MT423745.1 | 17802 | Brixen, Albanbühel | metatarsus distal | - | | Early BA | T3 |
| 25 | BRI17 | OEG17 | 863 | Brixen, Köstlan | M3 | - | | Late BA | neg |
| 26 | BRI18 | OEG18 | 843 | Brixen, Köstlan | molar | - | | Late BA | neg |
| 27 | BRI19 | OEG19 | 841 | Brixen, Köstlan | molar | 1122–940 | MAMS-23517 | Late BA | neg |
| 28 | BRI20 | OEG20 | 375 | Brixen, Köstlan | molar | - | | | neg |
| 29 | BRI21 | OEG21 | 403 | Brixen, Köstlan | molar | neg | | Late BA | neg |
| 30 | BRI22 | OEG22 | 364 | Brixen, Köstlan | molar | - | | Late BA | neg |
| 31 | BRI23 | OEG23 | 384 | Brixen, Köstlan | molar | - | | Late BA | neg |
| 32 | BRI24 | OEG24 | 382 | Brixen, Köstlan | molar | - | | Late BA | neg |
| 33 | BRI33 | OEG33 | 363 | Brixen, Köstlan | molar | - | | Late BA | neg |
| 34 | KLA34 | OEG34 | 362+368 | Klausen, Gufidaun-Plank | radius proximal | - | | Late Chalcolithic | neg |
| 35 | KLA35 | MT423740.1 | 372 | Klausen, Gufidaun-Plank | tibia proximal | - | | Late Chalcolithic | T3 |
| 36 | KLA37 | MT423746.1 | 451 | Klausen, Gufidaun-Plank | tibia | - | | Late Chalcolithic | T3 |
| 37 | PA1 | MT423762.1 | 73–155 421/D | Savognin-Padnal | M3 | 1450–1350/1300 | | Middle BA | T3 |
| 38 | PA2 | MT423749.1 | 73–93 22/B | Savognin-Padnal | M3 | 1350/1300–900/800 | | Late BA | T3 |
| 39 | PA3 | MT423753.1 | 75–23b 411/D | Savognin-Padnal | M3 | 1450–1350/1300 | | Middle BA | T3 |
| 40 | PA4 | MT423732.1 | 76–9b 54/E | Savognin-Padnal | M3 | 1950/1900–1550/1450 | | Early BA/ beginning Middle BA | T2 |
| 41 | PA6 | MH6 | 72–150 201/B | Savognin-Padnal | M3 | 1350/1300–900/800 | | Late BA | neg |
| 42 | PA7 | MT423733.1 | SP76/126 51 143/E | Savognin-Padnal | M3 | 1950/1900–1550/1450 | | Early BA/ beginning Middle BA | Q |
| 43 | PA8 | MT423750.1 | 55/C | Savognin-Padnal | M3 | 1450–1350/1300 | | Middle BA | T3 |

**Table 1.** *Cont.*

| | Sample Code | Genbank Acc. No./Lab Code | Archaeological Code/Horizon | Site | Element | Dating | 14 C Lab Code | Dating | MtDNA Haplogroup |
|---|---|---|---|---|---|---|---|---|---|
| 44 | PA9 | MT423752.1 | SP76/189/E | Savognin-Padnal | M3 | 1950/1900–1550/1450 | | Early BA/ beginning Middle BA | T3 |
| 45 | PA10 | MT423760.1 | SP72/120–23/B | Savognin-Padnal | M3 | 1350/1300–900/800 | | Late BA | T3 |
| 46 | PA11 | MH11 | 75–27c 411/D | Savognin-Padnal | M3 | 1450–1350/1300 | | Middle BA | neg |
| 47 | PA12 | MH12 | 56/C | Savognin-Padnal | M3 | 1450–1350/1300 | | Middle BA | neg |
| 48 | PA13 | MT423760.1 | SP73/102–22/B | Savognin-Padnal | M3 | 1350/1300–900/800 | | Late BA | T3 |
| 49 | PA14 | MT423761.1 | SP73/96–201/B | Savognin-Padnal | M3 | 1350/1300–900/800 | | Late BA | T3 |
| 50 | PA15 | MT423747.1 | SP76/9(b) HZ-54/E | Savognin-Padnal | M3 | 1950/1900–1450 BC | | Early BA/ beginning Middle BA | T3 |
| 51 | PA16 | MH16 | 73–151 22/B | Savognin-Padnal | M3 | 1350/1300–900/800 | | Late BA | neg |
| 52 | PA17 | MT423763.1 | SP73/150–201/B | Savognin-Padnal | M3 | 1350/1300–900/800 | | Late BA | T3 |
| 53 | PA18 | MT423764.1 | 75–141b 52/E | Savognin-Padnal | M3 | 1950/1900–1550/1450 | | Early BA/ beginning Middle BA | T3 |
| 54 | PA19 | MT423765.1 | SP75/166–411/D | Savognin-Padnal | M3 | 1450–1350/1300 | | Middle BA | T3 |
| 55 | PA20 | MT423734.1 | SP73/166–202/B | Savognin-Padnal | M3 | 1350/1300–900/800 | | Late BA | Q |
| 56 | PA22 | MT423748.1 | 75–20b 401/D | Savognin-Padnal | M3 | 1450–1350/1300 | | Middle BA | T3 |
| 57 | PA23 | MT423766.1 | SP73–108 202.1 (black)/B | Savognin-Padnal | M3 | 1350/1300–900/800 | | Late BA | T3 |
| 58 | PA24 | MT423751.1 | SP73–108 202.2 (red)/B | Savognin-Padnal | M3 | 1350/1300–900/800 | | Late BA | T3 |
| 59 | PA78 | MT423735.1 | | Savognin-Padnal | horncore sin | 1950/1900–1550/1450 | | Early BA/ beginning Middle BA | T2 |
| 60 | PA178 | MT423735.1 | | Savognin-Padnal | horncore dex | 1950/1900–1550/1450 | | Early BA/ beginning Middle BA | T2 |

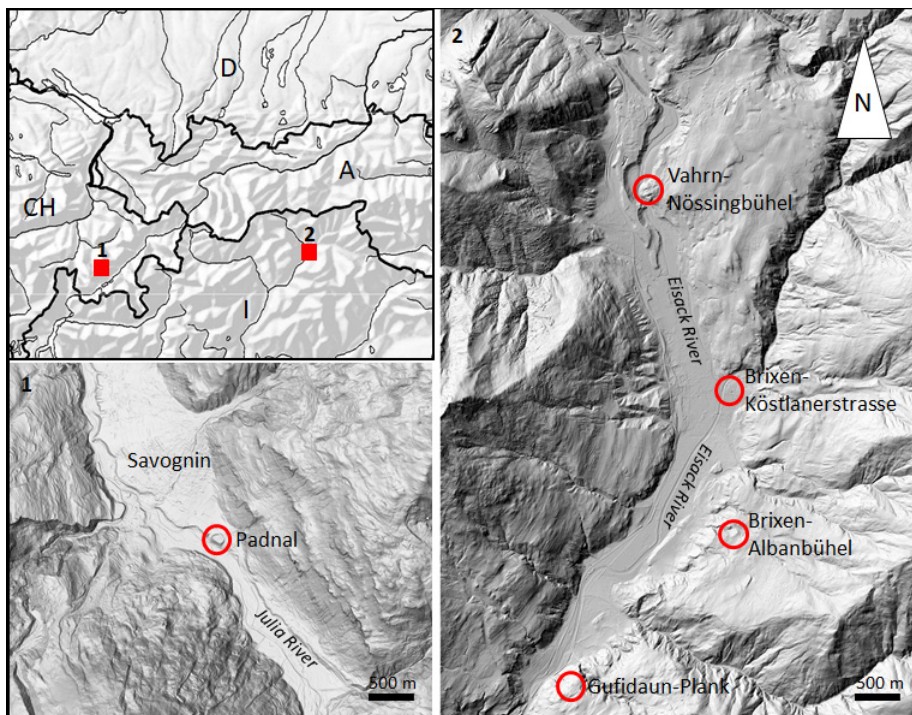

**Figure 1.** Topographical maps showing an overview and details of the locations of the sites in Switzerland (1) and South Tyrol, Northern Italy (2). ©swisstopo; Thomas Reitmaier Archaeological Service of the canton Grison, Switzerland, and Klaus Oeggl University of Innsbruck.

*2.2. The Archaeological Sites in Switzerland and Northern Italy*

2.2.1. Northern Italy, Brixen Albanbühel 46°37′20″ N 11°53′40″ E

This stone wall-ditch fortified farming site lies midway along the Eisack valley overlooking the eastern side of the Brixen (ital. Bressanone) basin at about 700 m above sea level. The first discoveries on the site were made between 1914 and 1915 during road works, with the construction of hairpin bends cutting into the hillside revealing the remains of a series of houses partly set into the slope, each with one or two living areas. These structural remains dated to various phases of the Bronze Age and to the Middle and Late Iron Age, respectively, as subsequent systematic excavations in the 1980s by the Cultural Heritage Office of Bolzano also proved. The settlement was predominantly occupied during the late Early Bronze Age and Middle Bronze Age [31–33]. With the Middle Bronze Age cattle represent >40% of the faunal remains, supporting their importance for the inhabitants (Figure 1).

2.2.2. Northern Italy, Brixen Köstlanerstrasse 46°42′51″ N 11°39′48″ E

Excavations were carried out in 2002–2003 in Brixen—Köstlanerstrasse by the Archaeological Heritage Office of Bolzano and unearthed a huge settlement area (2000 m²) dating to the Late (Final) Bronze Age (Laugen A) [31]. The site is located on an ancient river terrace of the river Rienz (ital. Rienza), near its confluence with the river Eisack (ital. Isarco), at about 560 m above sea level.

The investigated area shows two sectors: the first is characterized by settlement structures (layouts of houses, embankments, an embankment with probable defensive function and/or delimitation of the settlement), and the second sector, adjacent to the first, contains numerous pits of various shapes and sizes partly used as refuse chutes, partly as cooking pits (Figure 1).

### 2.2.3. Northern Italy, Vahrn Nössingbühel 46°45′0.94″ N, 11°38′38.12″ E

The fortified farming site of Nössing lies at 660 m above sea level (65 m above the level of the river Eisack) in the municipality of Vahrn, north of Brixen. The Early Bronze Age settlement occupies the summit of a shale hill, naturally fortified on three sides by steep precipices to the left bank of the river Eisack. The northern side, less naturally defended, has a defensive wall contemporary with the settlement. Research carried out by the University of Padua in the 1960s led to the discovery of a large body of material culture (pottery, flint tools, other types of worked stone, worked bone and antler, and metalwork) in addition to an important assemblage of faunal remains [32]. Some chronologically and culturally significant finds include a disc-headed pin with wire wound around the shank (Horkheimernadel), characteristic of the north-alpine Bronze Age A1b phase, and a fragmentary clay tablet (Brotlaibidol). In terms of material culture such as ceramics, the contribution of the north Italian early Bronze Age Culture Polada type in the broad sense is significant, while that of the North Alpine cultural groups (Oberrhein/Hochrhein and Singen groups, the Straubing Culture, and, subsequently, the Arbon Culture and Late Straubing) appears to be more important than in other contemporary contexts in the South Tyrol area [33] (Figure 1).

### 2.2.4. Northern Italy, Klausen Gufidaun-Plank: 46°38′54.65″ N, 11°35′55.46″ E

The archaeological site was discovered in 2005 during construction works and is located on the south-eastern outskirts of the village of Gufidaun (ital. Gudon) located above the small city of Klausen, on the left side of the Eisack river at about 600 m above sea level. The territory of Gufidaun consists of a gently sloping terrace, elevated above the river, and is well known for a continuity of the prehistoric, protohistoric, and historic age settlements. Apart from the Iron Age and Roman layers and structures, the most interesting aspect is a small terraced area in which abundant metal slag was found within at least two "furnaces" or stone structures for combustion activities and the presence of clay tuyeres, supporting the smelting of copper ore [34,35] (Figure 1).

### 2.2.5. Switzerland, Savognin Padnal 46 35′51.98″ N, 9°35′59.97″ E

The site of Savognin Padnal is located in the central alpine region of Grisons, Switzerland. The settlement was established at the beginning of the Bronze Age—around 1950/1900 BCE—and was apparently continuously inhabited until the end of the Late Bronze Age—around 900/800 BCE [30].

The settlement was established on a natural hill top, about 1210 m above sea level along the Julier alpine pass route [30]. Although traces of mining, bronze production and trading were found at the site, it is more likely that the people of Padnal primarily lived as farmers and pastoralists [19,30]. Based on 40,000 bone fragments, cattle was the most frequent species, and during the Bronze Age, they increasingly dominated the assemblage (over 60% in LBA), while the proportion of sheep and goat decreased over time (less than 30% in LBA). Furthermore, the cattle mortality profile also changed during the Middle and Late Bronze Age. The slaughter age rose over time which is a clear indication for secondary production, meaning that dairy and traction were more important aspects of cattle than pure meat supply [19] (Figure 1).

### 2.3. DNA Extraction, PCR, and Sequencing

Teeth, bones, and two horn cores were prepared and DNA was extracted twice or more independently as described by [36], except that $2 \times 100$ mg per sample was used. In case of the horn cores, 100 mg was used. At least two mock extraction controls were performed for each extraction series of 8 samples. The two extracts were combined and ultrapurified with buffer AE (provided from DNeasy® Blood & Tissue Kit, Qiagen, Basel, Switzerland) using 30 kD filter units (Amicon/Millipore, Zug, Switzerland). The final eluate was 100–200 μL.

PCR reactions are slightly modified from [37]. In particular: several partially overlapping targets of mitochondrial d-loop with different lengths covering nucleotide positions 15′928–16′271 of the reference sequence V00654 (without primers) were PCR amplified (Table S1). Amplification was carried out in 25 μL volumes. One reaction contained 1.5–2.5 U AmpliTaq Gold, 1× GeneAmp PCR Gold buffer (150 mM Tris-HCl, 500 mM KCl, pH 8.0) and 2 mM MgCl$_2$ (all Applied Biosystems, Hombrechtikon, Switzerland); 0.2 mM dNTP Mix (Promega, Dübendorf, Switzerland); 1 μM of each primer; and 5–10 μL template DNA on a Mastercycler ProS (Eppendorf, Allschwil, Switzerland). The cycling conditions were 11 min initial denaturation, followed by 70 cycles of denaturation at 95 °C for 1 min, annealing at 52–55 °C for 1 min and extension 72 °C for 1 min, with a final extension of 60 s (5 min for Padnal) at 72 °C. At least one non-template control was performed with a set of eight amplifications. PCR products were purified from 3% agarose, LE, analytical grade (Promega, Dübendorf, Switzerland) gel using MinElute Gel Extraction Kit (Qiagen, Hombrechtikon, CH). At least two PCR products per marker were directly Sanger sequenced forward and reverse by Microsynth (Balgach, Switzerland) using tailed PCR primer [38] (Table S1).

*2.4. Authenticity*

Established standards in aDNA research at Integrative Prehistory and Archaeological Science (IPAS) were adhered to [36,39] as described in [37]. Briefly: all ancient DNA work (pre-PCR) was performed in dedicated, physically separated laboratories, following a strict one-way policy. Benches and tools were treated with bleach and UV-irradiated, consumable plastic ware was UV-irradiated prior to use. No PCR products were observed in the negative controls. Each target was validated with two or three independent extractions and up to three PCR products per marker and extract.

Sequence data generated in this paper are available at Genbank with acc. numbers: MT423732–MT423770.

*2.5. Statistical Analyses*

Sequences were aligned with BioEdit version 7.0 (http://www.mbio.ncsu.edu/BioEdit/bioedit.html) and by eye, using the *Bos taurus* reference genome V00654. A total length of concatenated sequences of 320 bp were used for further analyses. Haplogroup identification was according to [26,40,41] and confirmed by a rooted ML tree performed with MEGA version X (Kumar, Stecher, Li, Knyaz, and Tamura 2018; data not shown).

A median joining network was built with NETWORK 4.6.1.2 [42]. Population diversity measures, F$_{st}$ values, mismatch analyses were calculated with Arlequin ver 3.5.2.2 [43] using pairwise distances and bootstrap set to 100.

Nonmetric multidimensional scaling (MDS) was computed with the PAST version 4 software [44] using Reynolds genetic distances calculated with Arlequin [43] based on a 197 bp sequence length shared between all used sequences. Modern cattle populations were broadly geographically grouped to Italy (n = 213), East (n = 94), West (n = 46), North (n = 22), Switzerland CH (n = 22), and Chalcolithic (n = 8) [26,45–48] (Table S2). Archaeological samples from Padnal (n = 19) and Albanbühel (n = 13) are according to Table 1.

## 3. Results

### 3.1. mtDNA D-Loop Haplotypes, Genetic Diversity, and Relationship between Haplotypes

A total of 60 cattle remains from five different archaeological sites in the alpine area from South Tyrol, Italy, and Switzerland were tested for PCR amplification of taurine mtDNA d-loop fragments. Moreover, 40 samples were successfully typed according to the criteria described in methods. Available sample size and amplification success varied between the sites, in particular all nine molars of the site Brixen Köstlan were negative, probably due to bad preservation conditions, as all samples were excavated from close to the surface (Table 1).

Based on 29 SNPs, 17 individuals belong to the T3 main haplotype, 13 are different haplotypes of haplogroup T3, another three belong to T3/T4, T4 being a subgroup of T3 [41]. We detected five cattle belonging to four haplotypes of the T2 haplogroup and two cattle affiliate to the Q haplotype (Figure S1). The two left and right horn cores PA78 and PA178 carry the identical T2 haplotype and probably belong to the same animal; this interpretation is supported by the fact, that they belong to the same layer.

Within the dataset, members of haplogroup T2 were detected earliest in Early Bronze Age and of haplogroup Q at Early and Late Bronze Age.

Haplotype (HTdiv) and genetic diversity (MNPD, π) were high at the sites Albanbühel and Padnal. The five cattle from Varn-Nössingbühel have zero diversity, all animals belong to the main T3 haplotype (Table 2). Although based on a limited number of individuals, the data indicate a severe bottleneck in this population. The site Gufidaun Plank was excluded because only two individuals were available.

**Table 2.** Population diversity measures and demographic statistics from the sites Albanbühel, Savognin Padnal, and Nössingbühel. * denotes significant P values on 0.05 level.

| Parameter | Albanbühel | Padnal | Nössingbühel |
|---|---|---|---|
| N | 13 | 19 | 5 |
| Htdiv | 0.85 ± 0.088 | 0.83 ± 0.086 | 0 |
| MNPD (π) | 2.05 ± 1.22 | 2.42 ± 1.37 | 0 |
| Tajima's D | −1.45 | −2.01 | nd |
| Tajima's D *p*-value | 0.07 | 0.01 * | nd |
| Fu's Fs | −2.21 | −6.87 | nd |
| Fu's Fs *p*-value | 0.057 | 0.0 * | nd |
| Sudden expansion model: Harpending's raggedness index | 0.064 | 0.29 | nd |
| Harpending's raggedness index *p*-value | 0.49 | 0.8 | nd |
| Spatial expansion model: Harpending's raggedness index | 0.06 | 0.029 | nd |
| Harpending's raggedness index *p*-value | 0.69 | 0.84 | nd |

The median joining network shows that almost all haplotypes are private, except the main T3 haplotype, the Q haplotype shared by two individuals from Padnal (PA20, PA7), and the T3/T4 haplotype shared by two individuals from Albanbühel (ALB31, ALB28) (Figure 2).

*3.2. Population Demographic Estimates*

Keeping in mind that archaeological samples are not a real intermixing population, to tentatively infer demographic events such as the sudden increase in population size or spatial expansion, mismatch analyses of cattle from the sites Albanbühel, North Italy, and Savognin Padnal, Switzerland, were performed. The site Vahrn-Nössingbühel was not considered, because of very low sample size of five individuals.

Tajima's D and Fu's Fs are slightly negative with statistical significance for the Padnal population and supportive of either a demographic expansion, or best explained as an admixture event in the past by incoming cattle. The mismatch graph shows corresponding slightly bimodal patterns at both sites (Figure 3), which is more pronounced at Albanbühel. Non-significant raggedness indices show support for either recent demographic or spatial expansion (Table 2). The slightly bimodal mismatch is not the result of haplogroups Q or T2, because they are not very divergent to T3.

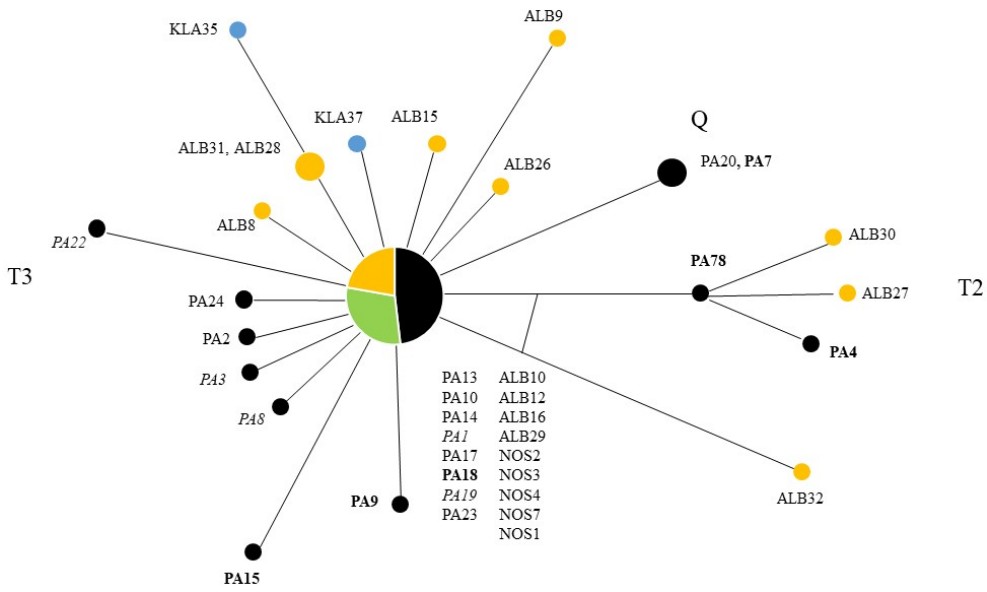

**Figure 2.** Median joining network of 320 bp mtDNA d-loop of alpine cattle, mutational positions are given according to the V00654 reference sequence. Black: Savognin Padnal (Bronze Age), orange: Albanbühel (Middle Bronze Age), green: Vahrn-Nössingbühel (Early Bronze Age), blue: Klausen Gufidaun Plank (Late Chalcolithic). Archaeological samples from this study are according to Table 1. Early Bronze Age samples from Savognin Padnal are written in bold, Middle Bronze Age in italic, Late Bronze age in normal letters.

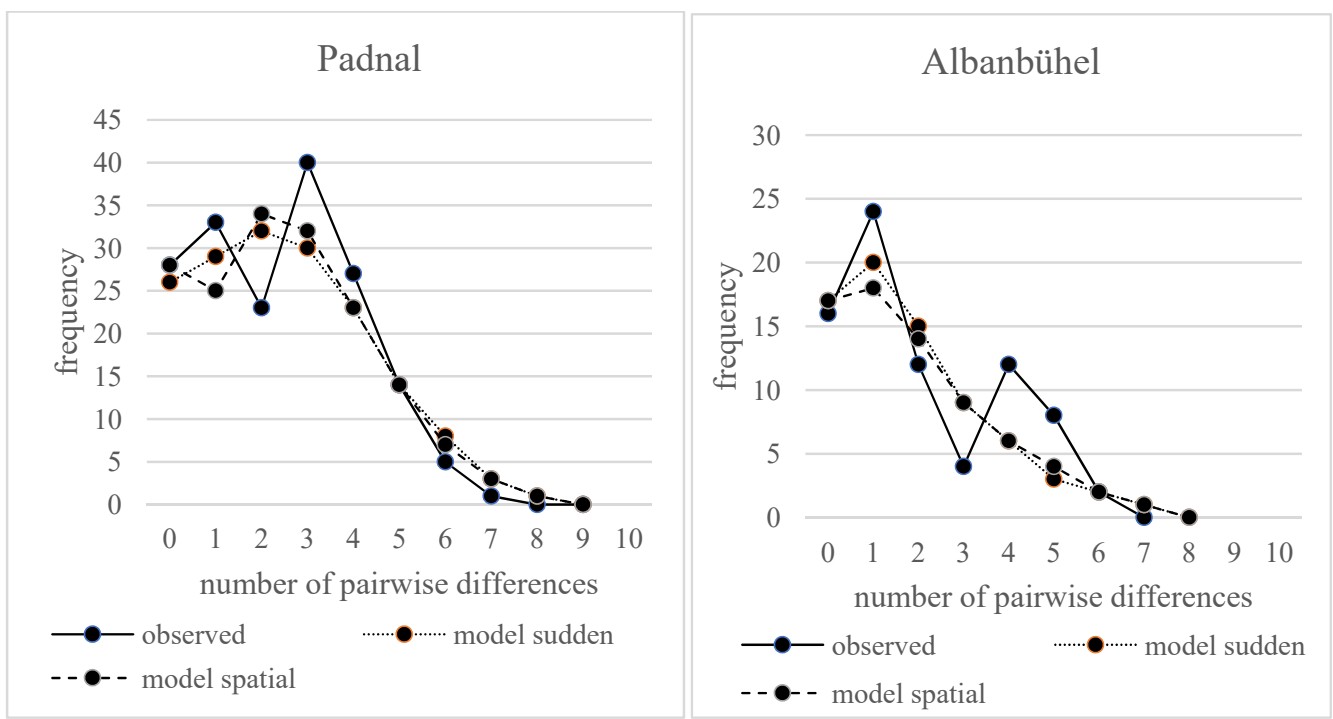

**Figure 3.** Mismatch distribution of cattle from the sites Savognin Padnal and Albanbühel.

The $F_{st}$ value of $-0.013$ (a negative $F_{st}$ value corresponds to zero, $p = 0.59$) is not significant and indicates no structuring between both populations.

### 3.3. Chronological Changes in Diversity in Cattle from Savognin Padnal

Although based on a limited number of individuals per time slot a trend was observed: genetic diversity is very high in Early Bronze Age and decreases about 4× from Early Bronze Age to Late Bronze Age (Table 3). T2 and Q contributed diversity to Early and Late Bronze Age cattle, where five of the eight cattle belong to the main T3 haplotype.

**Table 3.** Chronological development of cattle diversity at Savognin Padnal.

| Period | N | Htdiv | MNDP ($\pi$) |
|---|---|---|---|
| Early Bronze Age | 6 | 1 ± 0.009 | 4.33 ± 2.49 |
| Middle Bronze Age | 5 | 0.9 ± 0.16 | 1.6 ± 1.13 |
| Late Bronze Age | 8 | 0.64 ± 0.18 | 1.25 ± 0.88 |

### 3.4. Bronze Age Cattle Populations in Relation to Modern Cattle Breeds

Non-metrical MDS was chosen to explore relationships between Bronze Age cattle and modern rural breeds in Europe. Based on genetic distances, both ancient populations do not cluster together but are different. Grouped into geographical groups, cattle from Albanbühel and Padnal are linked to Italian cattle, Padnal is also close to Swiss cattle (Figure 4). The very low stress factor of 0.033 indicates an excellent representation of the data within the two dimensions. Looking at individual breeds, Bronze Age cattle are placed within different Italian, Swiss, and western breeds, but to only very few eastern European breeds and to none of the two northern breeds. The stress factor indicates a sufficiently good representation (Figure S2). The topology of the MDS plot is further supported by the close placement of two different datasets from Italian Maremmana cattle (breed 5 and 12).

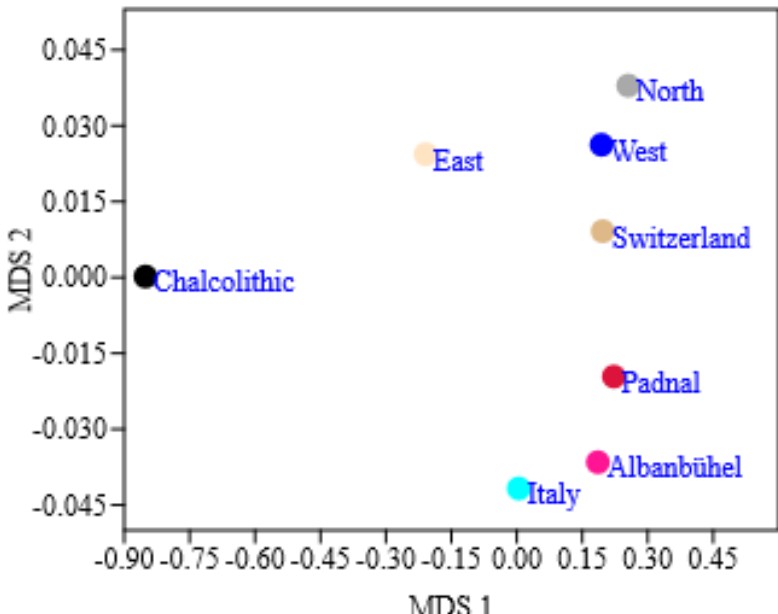

**Figure 4.** Non-metric MDS plot based on Reynolds genetic distances and 197 bp d-loop sequences from cattle breeds broadly grouped to different geographic regions (the geographic grouping of breeds is given in Table S2).

### 4. Discussion

Domestic animals such as cattle are valuable proxies for human actions, and their joint past can be traced through ancient DNA, e.g., [49–51]. Domestic animal remains are generally more frequently found in archaeological contexts than human remains, making them crucial evidence to understand past societies. Here we exploit partial mtDNA d-loop data from 40 alpine cattle from different localities during the Bronze Age, a time period

characterized by profound technological and agricultural innovations and increased human migrations and trade networks [3,4,52]. Concerning the Alpine region, contact appears more difficult, yet longtime transfer/exchange routes across the alps existed with, e.g., Savognin Padnal being one of the tipping points [4]. The same is true also for South Tyrol in northern Italy [7,16,53].

The two sites from South Tyrol are both fortified; however, non-contemporaneous settlements with Nössingbühel dating to the Early Bronze Age and Albanbühel from the Late Early to Middle Bronze Age as well as the cattle show very different genetic patterns: All five Nössingbühel cattle belong to the same main T3 haplotype, suggesting a strong bottleneck in the maternal lineage. This is not caused by isolation of the site as the material culture support contact to northern Italy and to a lesser extend to the north-west [33]. Could it be a disease, selection, or inbreeding within a small population kept in one area, as seen in Chillingham cattle today [54]? Archaeozoological data support the use of cattle for meat at Nössingbühel where it was well possible to farm and pasture [33]. Although we cannot exclude that this finding results from low sample size, the identification of two diverse haplotypes within two samples for cattle from Gufidaun Plank indicates the result is not due to a sample bias. On the contrary, cattle from Albanbühel are highly diverse (Table 3, Figure 4) and show evidence for population expansion and/or admixture events. These events can predate the Bronze Age, e.g., linked to the spread of agriculture. However, in the MDS plot cattle from Albanbühel are placed far away from Chalcolithic cattle from the East [26] and also from Eastern European rural breeds indicating no link to Eastern Europe as expected for this hypothesis. On the other side, Albanbühel cattle have close ties to Italian rural breeds, which may indicate intensive contact and exchange with the south during the Early Bronze Age along the natural passages through the Alps in a roughly north-south direction. It is plausible that Albanbühel cattle reflect incoming new cattle from the south; however, other data, e.g., genomic data or particularly isotopes, will be necessary to substantiate this hypothesis.

Similar to Albanbühel, the cattle from Savognin Padnal—a farming site with metallurgical activities—in a distant geographical area have genetically diverse maternal lineages and exhibit patterns for expansion/admixture. In the MDS plot Padnal is positioned close to Albanbühel and Italian cattle with some ties to Swiss rural cattle and no link to Chalcolithic and Eastern European cattle. Bopp-Ito (2019) hypothesized that the small cattle at Padnal were brought in from Italy [55]. This makes sense as the site is located on the route to the mountain pass Julier and could have been an important junction of cultural exchange [4], including for livestock. Interestingly, genetic diversity declines about four-fold between the Early and Late Bronze Age and haplotype diversity is also reduced. This may reflect changes in cattle use towards milk or traction as evidenced by high slaughter age, decreasing population size because less cattle were needed for food or reduced gene flow starting between Early and Middle Bronze Age, as is seen also in pigs [56] at Savognin Padnal.

However, because of the presence of T2 and Q haplotypes, the alpine populations of Padnal (T2 and Q) and Albanbühel (T2) may instead be related to cattle from a wider area. For T2 one option is the Balkans, where today many breeds have high proportions of T2 [27,29], South-Western or Central Europe, where T2 is frequently found [26]. For Q another link to northern Italy where this haplotype is frequent is possible [40]. It is also noteworthy that Q was very frequent in South-East Europe, making up 50% of individuals since the Neolithic.

Regrettably there is very little other information on Bronze Age cattle populations available for European sites or for the alpine region.

To better understand issues around cattle husbandry, import, and transhumance in the past, short d-loop mtDNA data are less informative, as they represent long-term female histories of individuals; genomic NGS data, such as targeted amplicon sequencing will generate more detailed information about the Alpine cattle. Other limiting issues are the few sites and the rather broad dating periods, the latter problem only to be overcome at

sites with dendrochronological dating. Currently, isotopic and archaeozoological data from the sites of South Tyrol are being evaluated and will add further details to the complex history of alpine animal management.

**Supplementary Materials:** The following are available online at https://www.mdpi.com/article/10.3390/d13090449/s1, Figure S1: Observed SNPs in 320 bp of mtDNA d-loop fragments of cattle from the sites Savognin Padnal (PA), Brixen Albanbühel (ALB), Vahrn-Nössingbühel (NOS), Klausen Gufidaun-Plank (KLA). Figure S2: MDS plot based on Reynolds geneticdistances obtained from different cattle breeds. Table S1: Primer for PCR and sequencing used in this study. Table S2: Sequences used for genetic diversity measures and MDS Plot.

**Author Contributions:** Conceptualization, K.O., J.S. and A.S.; methodology, A.S., J.G. and M.H.; writing—original draft preparation, A.S.; writing—review and editing, J.G., M.H., K.O., U.T., J.S. and A.S.; funding acquisition, K.O., J.S. and U.T. The genetic results of the site Savognin Padnal were the MSc thesis of M.H. All authors have read and agreed to the published version of the manuscript.

**Funding:** This research was conducted in the framework of the special research program HiMAT (The History of Mining Activities in the Tyrol and Adjacent Areas—Impact on Environment & Human Societies) at the University of Innsbruck and funded by the Autonome Provinz Bozen -Südtirol, Abteilung Bildungsförderung, Universität und Forschung (grant no. 14/40.3 from 6 December 2012) and the Austrian Science Fund FWF (grant no.: F3111-G02).

**Institutional Review Board Statement:** Not applicable.

**Informed Consent Statement:** Not applicable.

**Data Availability Statement:** Genbank with acc. numbers: MT423732–MT423770.

**Acknowledgments:** We heartily thank the team HiMAT for inspiring discussions during our meetings, Miki Bopp-Ito for sharing her knowledge about the animals from Savognin Padnal, Thomas Reitmaier, the Federal Office of Topography swisstopo and Amanda Zwicky, Archäologischer Dienst Graubünden, for the supply of the geospatial data of the study area in Grison and Amt für Bodendenkmäler der Autonomen Provinz Bozen Südtirol (director Catrin Marzoli) for the data from South Tyrol. Last but not least many thanks to Lizzie Wright to make the best of our English.

**Conflicts of Interest:** The authors declare no conflict of interest.

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
