# Peer review of "MtDNA D-Loop Diversity in Alpine Cattle during the Bronze Age"

_diversity, doi:10.3390/d13090449_

Round 1

Reviewer 1 Report

In the submitted manuscript entitled “MtDNA d-loop Diversity in Alpine Cattle during the Bronze Age” Granado and colleagues try to reconstruct the changes of the cattle populations in the Alps during the Bronze Age. To do that they studied a small part of the maternal uniparental system, the mitochondrial DNA, in 40 samples (from 60 ancient remains), extracted from five Alpine archaeological sites between Italy and Switzerland and dated between the 2100 and 800 BCE.

The authors tried to answer different questions; the first is the classification of the animals in the cattle mtDNA haplogroups, then they tried to infer information about demographic changes in the population during the Bronze Age, and finally they checked for possible relationship with modern cattle populations in Europe.

Although the results are based on a small part of mtDNA and with few samples, they were obtained with standard and verified methods, so they can be considered valid. For these reasons I suggest to accept the manuscript for publication, after having solved some minor points listed below:

  • It’s important to write in the abstract the number of animal data generated for this work, as well as the fact that the authors analysed only partial D-loop data
  • Table 1 is not completely visible in the PDF file, please correct and make all the tables in the paper in a concordant form. Moreover, it is better not to use Christian references in dating, so instead of BC and AD it would be better to standardize everything to BCE (Before the Common Era).
  • I think that the 2.5 chapter should be moved above chapter 2.4.
  • I don’t understand why PA78 and PA178 are the same individual, is it just because they share 320bp on the mtDNA? Please explain.
  • Row 233: “We detected 5 cattle belonging to four variants of the T2 haplogroup and two cattle affiliate to the Q”. The term “variants” is usually associated with SNPs and not lineages, so I suggest to rephrase this sentence to avoid misunderstanding.
  • The proposed network is interesting, but it would give more phylogenetic information if you use a rooted network adding an outgroup (for example an mtDNA from the P or R haplogroup). In addition, I would add information about the age of the samples, in the three groups used in Table 1, maybe using different shapes in order to corroborate the assumption in chapter 3.3. Please correct the overlap between the name of PA13 and the variant in the PA24 branch.
  • Please add the axes titles in Figure 3.
  • In chapter 3.3, the authors suggest a decrease in the variability between Early BA and Late BA at the Savognin Padnal site, suggested by the disappearance of haplogroups other than T3 and by the sharing of the same haplotype within the T3 haplogroup in the Late BA. First of all, in Table 1 the sample PA20 is affiliated to haplogroup Q and is classified as Late BA (please check or rewrite the sentence). In addition, I would like to know if the fact that 5 animals share the same haplotype cannot be a bias of the sampling (same family), since in the part related to archaeological sites (2.2) there are no details about it.
  • In the MDS plot (Figure 4) please specified Europe in East, North and West. Moreover, it could be interesting to see also a plot with the Pandal animals separated in the three time groups. In the related Figure S2, it is written that Albanbühel (29) and Padnal (28) are in red, but they are actually pink and purple. Moreover compared to Figure 4 the colours between the two sites are inverted. Please keep the colours in agreement as well as the axis titles between the two images.
  • Also in the conclusions you should say that the analyses are based on a part of the D-loop (Row 299) and include the number of final sequences analyzed.
  • What does the question marks (row199-202) mean in Table S2? Please separate the GeneBank ID from the sample name in the column B.

Reviewer 2 Report

This is an interesting paper that describes genetic diversity in ancient Alpine cattle. However, I suggest that the manuscript needs to be improved before consideration for publication.

My concerns are following:

  1. The caption to the figures should be under the figures
  2. In the Table 1 instead of Lab code, Genbank accession numbers should better be given.
  3. Quality of the Figure2 should be improved. In the present form it is not readable
  4. The alignment method should be indicated (i.e. MUSCLE, Clustal, etc.).
  5. The authors did not describe how the haplogroups were determined.
  6. Line 289 – PCA or MDS? Same on line 320

Round 2

Reviewer 2 Report

The manuscript has been improved according to the suggestions of reviewers. But I still have some comments:

line 204 - should be "Haplogroup identification..." instead of "Haplotype identification..." 
line 207 - should be "Arlequin ver 3.5.2.2" instead of "Arlequin WinARL35"
line 210 (also lines 299 and 362) - should be "Reynolds genetic distance" instead of "Reynold’s pairwise Fst". And the citation for this distance is needed: https://doi.org/10.1093/genetics/105.3.767
